# Policy Response, Social Media and Science Journalism for the Sustainability of the Public Health System Amid the COVID-19 Outbreak: The Vietnam Lessons

**Viet-Phuong La** [1,2], **Thanh-Hang Pham** [3,4,*], **Manh-Toan Ho** [1,2,*], **Minh-Hoang Nguyen** [1,2], **Khanh-Linh P. Nguyen** [1,2], **Thu-Trang Vuong** [5], **Hong-Kong T. Nguyen** [1,2], **Trung Tran** [6], **Quy Khuc** [7], **Manh-Tung Ho** [1,2] and **Quan-Hoang Vuong** [1,8,*]

1   Centre for Interdisciplinary Social Research, Phenikaa University, Yen Nghia Ward, Ha Dong District, Hanoi 100803, Vietnam; phuong.laviet@phenikaa-uni.edu.vn (V.-P.L.); hoang.nguyenminh@phenikaa-uni.edu.vn (M.-H.N.); linh.nguyenphuckhanh@phenikaa-uni.edu.vn (K.-L.P.N.); htn2107@caa.columbia.edu (H.-K.T.N.); tung.homanh@phenikaa-uni.edu.vn (M.-T.H.)

2   A.I. for Social Data Lab, Vuong & Associates, 3/161 Thinh Quang, Dong Da District, Hanoi 100000, Vietnam

3   Faculty of Management and Tourism, Hanoi University, Km9, Nguyen Trai Road, Thanh Xuan, Hanoi 100803, Vietnam

4   School of Business, RMIT Vietnam University, Hanoi 100000, Viet Nam

5   Sciences Po Paris, 27 Rue Saint-Guillaume, 75007 Paris, France; thutrang.vuong@sciencespo.fr

6   Vietnam Academy for Ethnic Minorities, Hanoi 100000, Vietnam; trungt1978@gmail.com

7   Faculty of Economics and Finance, Phenikaa University, Yen Nghia Ward, Ha Dong District, Hanoi 100803, Vietnam; quy.khucvan@phenikaa-uni.edu.vn

8   Centre Emile Bernheim, Université Libre de Bruxelles, 1050 Brussels, Belgium

*   Correspondence: hangpt@hanu.edu.vn (T.-H.P.); toan.homanh@phenikaa-uni.edu.vn (M.-T.H.); hoang.vuongquan@phenikaa-uni.edu.vn or qvuong@ulb.ac.be (Q.-H.V.)

**Abstract:** Having geographical proximity and a high volume of trade with China, the first country to record an outbreak of the new Coronavirus disease (COVID-19), Vietnam was expected to have a high risk of transmission. However, as of 4 April 2020, in comparison to attempts to containing the disease around the world, responses from Vietnam are seen as prompt and effective in protecting the interests of its citizens, with 239 confirmed cases and no fatalities. This study analyzes the situation in terms of Vietnam's policy response, social media and science journalism. A self-made web crawl engine was used to scan and collect official media news related to COVID-19 between the beginning of January and April 4, yielding a comprehensive dataset of 14,952 news items. The findings shed light on how Vietnam—despite being under-resourced—has demonstrated political readiness to combat the emerging pandemic since the earliest days. Timely communication on any developments of the outbreak from the government and the media, combined with up-to-date research on the new virus by the Vietnamese science community, have altogether provided reliable sources of information. By emphasizing the need for immediate and genuine cooperation between government, civil society and private individuals, the case study offers valuable lessons for other nations concerning not only the concurrent fight against the COVID-19 pandemic but also the overall responses to a public health crisis.

**Keywords:** coronavirus; COVID-19; SARS-CoV-2; pandemic; policy response; social media; science journalism; public health system; Vietnam

## 1. Introduction

> "Dans les champs de l'observation le hasard ne favorise que les esprits préparés."
> —Louis Pasteur (1822–1895)

As China grappled to contain the outbreak of the novel coronavirus disease (COVID-19), caused by the virus officially named Severe Acute Respiratory Syndrome Coronavirus 2 (SARS-CoV-2), in the first few months of 2020, Vietnam, which shares a border of 1281 kilometers and a high volume of trade with the northern giant [1], was bracing itself for a high risk of cross-border infections. Within more than two months from January 23 when the first case of COVID-19 was detected in Vietnam, there have been 239 confirmed cases with zero deaths [2]. During the same period, the number of infections in China had skyrocketed from 600 people, with 17 deaths to 82,526 cases with 3330 deaths [3].

Despite the differences in their domestic contexts, such abysmal contrast between the two neighbors could raise questions as to how Vietnam, a populous but less-resourced nation of nearly 100 million people, has managed to contain the spread of the new disease. This feat merits in-depth studies, especially in light of the World Health Organization's (WHO) declaration of COVID-19 as a pandemic on March 11th [4] and the chaotic self-quarantine and lockdown in various countries in Europe and America.

Vietnam was among the first countries to have confirmed cases of COVID-19, with the first two patients (both Chinese) detected on January 23 [5]. This study identifies four periods of disease outbreak in Vietnam, namely (i) pre-January 23, (ii) between January 23 and February 26 when the first batch of sixteen patients were tested and treated till their discharge, (iii) between February 27 and March 5 when there was no new case and (iv) and post-March 6 when the 17th patient was detected and led a new wave of infections from incoming tourists and returning travelers [6].

As Figure 1 shows, compared to other countries, the infection rate in Vietnam was evidently much lower than that in China, Italy, South Korea, the United Kingdom and the United States. All these countries, except China, had the first cases announced in January [3]. In the period from January 23 to February 25, the rise in cases in Vietnam was comparable to that in the United States, United Kingdom and Germany. From late February to March 5, the situation in Vietnam appeared under control with no new cases while cases in South Korea and Italy soared to 5766 and 3089, respectively [3]. From March 6 onward, the steep upward trend was seen in much of Europe and the United States, which as of March 30 was ranked first with 122,653 cases [3].

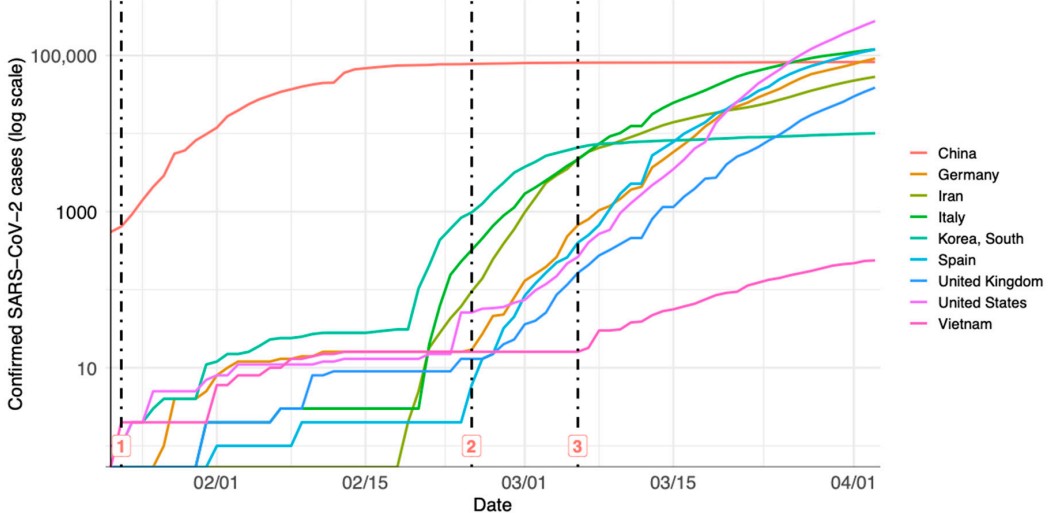

**Figure 1.** The rise of new Coronavirus disease (COVID-19) cases in selected countries (as of April 4, 2020). Note: ① is the first case in Vietnam; ② is when the patient 16 recovered, ended the 1st outbreak; ③ is the 17th case in Vietnam, also the beginning of the 2nd phase of the outbreak.

While Vietnam also saw an uptick in new cases during March, its response to COVID-19, which is a combination of political readiness, timely communication and scientific journalism, offers valuable lessons in dealing with situations of epidemics on a state level. As of April 4, the number of COVID-19 infections in Vietnam was 239, in which 90 patients had recovered, 149 are being treated and no deaths recorded (See Figure 1).

The study, though in its preliminary and subject to changes as the disease progresses, may nonetheless be instructive and helpful for other countries to better understand the role of policy response, social media and science journalism in maintaining public health. The case of Vietnam provides empirical evidence for assessing the efficacy of specific measures in fighting the pandemic.

## 2. Literature Review

Within the first three months of the new decade, the novel coronavirus, officially named SARS-CoV-2—and the corresponding disease COVID-19—has spread from Wuhan city in China's central province of Hubei to 201 other countries and territories. Over 1,123,024 people have been infected with over 59,140 lives lost as of April 4, 2020, according to the Global COVID-19 Tracker Map at John Hopkins University [7]. The rapid contagion and severity of the new disease has prompted WHO to update its statement, from classifying the outbreak as a "Public Health Emergency of International Concern" on January 30 to a "Pandemic" on March 11 [4].

Given the urgency of this outbreak, the international academic community is mobilizing ways to accelerate the development of disease detection and intervention. A statement by the research-charity based Wellcome Trust in London has gathered more than 100 signatories to ensure access to data and research findings on the disease could better inform the public and save lives. These include leading publishers such as Springer Nature, Elsevier or Taylor and Francis as well as prestigious journals such as *The Journal of the American Medical Association*, *The British Medical Journal*, the *Lancet* and *New England Journal of Medicine* [8]. In these leading journals, editorials echoed a call for researchers to "keep sharing, stay open" [9]. In *Nature Medicine*, the editorial also stated that "communication, collaboration and cooperation can stop the 2019 coronavirus" [10]. Editors in the leading medical journal BMJ asserted that "while scientists and public health professionals are working non-stop to contain the novel coronavirus, political scientists, economists and sociologists should also ready themselves for rapid response" [11].

Complementing the clinical research on COVID-19 are studies that integrate social sciences in the outbreak response. Social sciences research is expected to produce rich and detailed insights into the social, behavioral and contextual aspects of the communities, societies and populations affected by infectious disease epidemics. The overarching aim is to bring together social sciences knowledge and biomedical understanding of the COVID-19 epidemic. Such connection would strengthen the response at international, regional, national and local levels to stop the spread of COVID-19 and mitigate its social and economic impacts [12].

Among countries affected by the pandemic, Vietnam, with its geographical proximity with China, faces a high likelihood of being severely affected by the spread of the disease. Moreover, although the Vietnamese healthcare system is under-resourced and has inherent weaknesses [13–17], especially concerning health insurance and patient welfare [17,18], the Vietnamese response to urgent situations has been commendable. It is likely that the Vietnamese government has learnt from its experiences in the past, especially in dealing with the Severe Acute Respiratory Syndrome (SARS) epidemic in 2003. Vietnam's success in the effective control of SARS for the first time in the world was achieved by "complete isolation of patients and implementation of nosocomial infection control from an early stage of epidemic" [19]. The lesson is clear for Vietnam: early risk management requires taking adequate actions from the early stage of the disease.

The few studies on the outbreak in Vietnam have largely focused on the clinical aspects [20], with the exception of [21], which is on public risk perception. This piece analyzes the government's response in terms of public health measures and policy implementation, as well as the mobilization of

citizens' collaboration in containing the disease have been very limited. This shall be a subtle call to action for researchers in Vietnamese social sciences.

## 3. Materials and Methods

This paper reviews Vietnamese policy response, news and science journalisms related to COVID-19 recently. Findings were derived from the analysis of a database of recent policies, articles and the credibility of data sources in Vietnam. Extensive coverage was given to the pandemic in both the official press and academic journals as well as through reports, briefs and presentations by members of concerned organizations (e.g., WHO).

A Python-powered web crawler engine was used to scan the data from online newspapers in Vietnam, such as *Tuổi Trẻ, Thanh Niên, VnExpress* or *Kênh 14,* to name a few. Then, the scanned data were saved into a news analysis system, which is developed by.NET Core, for storage and future analysis. The data structure contains three main components:

- Projects & Data Sources: Settings for projects and news sources.
- Data Logging: Log of the data collection process
- News & Filters: Collected news with filters.

Examples of Python code are as follows (Figure 2):

```python
def parseDetail(self, response):
    projectId = response.meta.get('projectId')
    projectSourceId = response.meta.get('projectSourceId')
    sourceId = response.meta.get('sourceId')
    sourceUrl = response.meta.get('sourceUrl')
    projectCfg = response.meta.get('projectCfg')
    crawlCfg = response.meta.get('crawlCfg')
    parseCfg = response.meta.get('parseCfg')
    pageId = response.meta.get('pageId')

    keyGroup = response.meta.get('keyGroup')

    if self.existNewsUrl(projectId, response.url):
        return(None)

    #print(keyGroup)
    newsTitle = NewsParser.extractText(NewsParser.parseItem(response, parseCfg["TitleCfg"]))
    print(newsTitle)

    newsDate = NewsParser.parseDate(response, parseCfg["DateCfg"])
    print(newsDate)

    if projectCfg["UpToDate"]:
        upToDate = datetime.strptime(projectCfg["UpToDate"], "%Y-%m-%dT%H:%M:%S")
        if upToDate > newsDate:
            if projectSourceId not in self.completed:
                self.completed.append(projectSourceId)
                return(None)

    newsSapo = ""
    if "SapoCfg" in parseCfg:
        parseSapo = NewsParser.parseItem(response, parseCfg["SapoCfg"])
        newsSapo = NewsParser.extractText(parseSapo)
        #print(newsSapo)
```

**Figure 2.** Examples of Python code.

Using this system, we can set up sources and keywords. Furthermore, all tools and datasets will be maintained for future mining. We expect the dataset to keep growing over time, presenting us with new opportunities to extract deeper and more valuable insights.

In this article, using five keywords related to COVID-19, namely: *covid, ncov, corona, viem phoi* (Vietnamese for pneumonia, which has some symptoms in common with COVID-19), *sars-cov,* between January 9 and April 4, the tool has collected 14,952 news reports on the topic of concern, as presented in Table 1.

**Table 1.** List of online news sources (as of April 4, 2020).

| Sources | Start Date | News |
|---------|-----------|------|
| *kenh14.vn* | 15/01/2020 | 2132 |
| *vtc.vn* | 17/01/2020 | 909 |
| *suckhoedoisong.vn* | 09/01/2020 | 929 |
| *cafef.vn* | 12/01/2020 | 804 |
| *tuoitre.vn* | 12/01/2020 | 1196 |
| *chinhphu.vn* | 09/01/2020 | 441 |
| *Zing.vn* | 17/01/2020 | 1360 |
| *dantri.com.vn* | 10/02/2020 | 1838 |
| *plo.vn* | 23/01/2020 | 400 |
| *vnexpress.net* | 23/01/2020 | 917 |
| *vov.vn* | 27/02/2020 | 1748 |
| *nld.vn* | 28/02/2020 | 577 |
| *rfa.org* | 10/02/2020 | 248 |
| *thanhnien.vn* | 18/02/2020 | 1453 |

Raw data were manually cleaned then categorized based on its characteristics, such as the timeline of COVID-19 cases, the timeline of international events regarding COVID-19, media reports and policy response from the Vietnamese government. Regarding the social media aspect, due to technical limitations, we could not scan information from Facebook. Thus, we used the remediation of social media on news outlets as a proxy to explore the social media aspect. Keywords that uniquely fit with the aspect, including '*mạng xã hội*' (social media), '*cư dân mạng*' (netizens), Facebook and Zalo, were used to search within the collected dataset. Furthermore, the data of the VN INDEX, which represents the changes in the Vietnamese stock market's prices during the COVID-19 pandemic, was also added to complete the dataset. Finally, we store the cleaned data as a comprehensive dataset in excel files.

The dataset (updated as of April 4, 2020) is available at Open Science Framework (OSF) (URL: https://osf.io/4w9ef/; DOI: 10.17605/OSF.IO/4W9EF) [22]. Having organized the dataset, we then calculated descriptive statistics to illustrate how the Vietnamese government, news and science journalism respond to COVID-19.

## 4. Results

### 4.1. Chronology

Figure 3 presents a timeline of the spread of COVID-19 in Vietnam, tracing from the first identified patient in on January 23 to the most recent case on March 31. At this time, the number of cases in Vietnam exceeded 200 and for the sake of better presentation of data, we decided to cut off the data on this day. The most updated version of the chronology as well other data can be accessed from OSF [22].

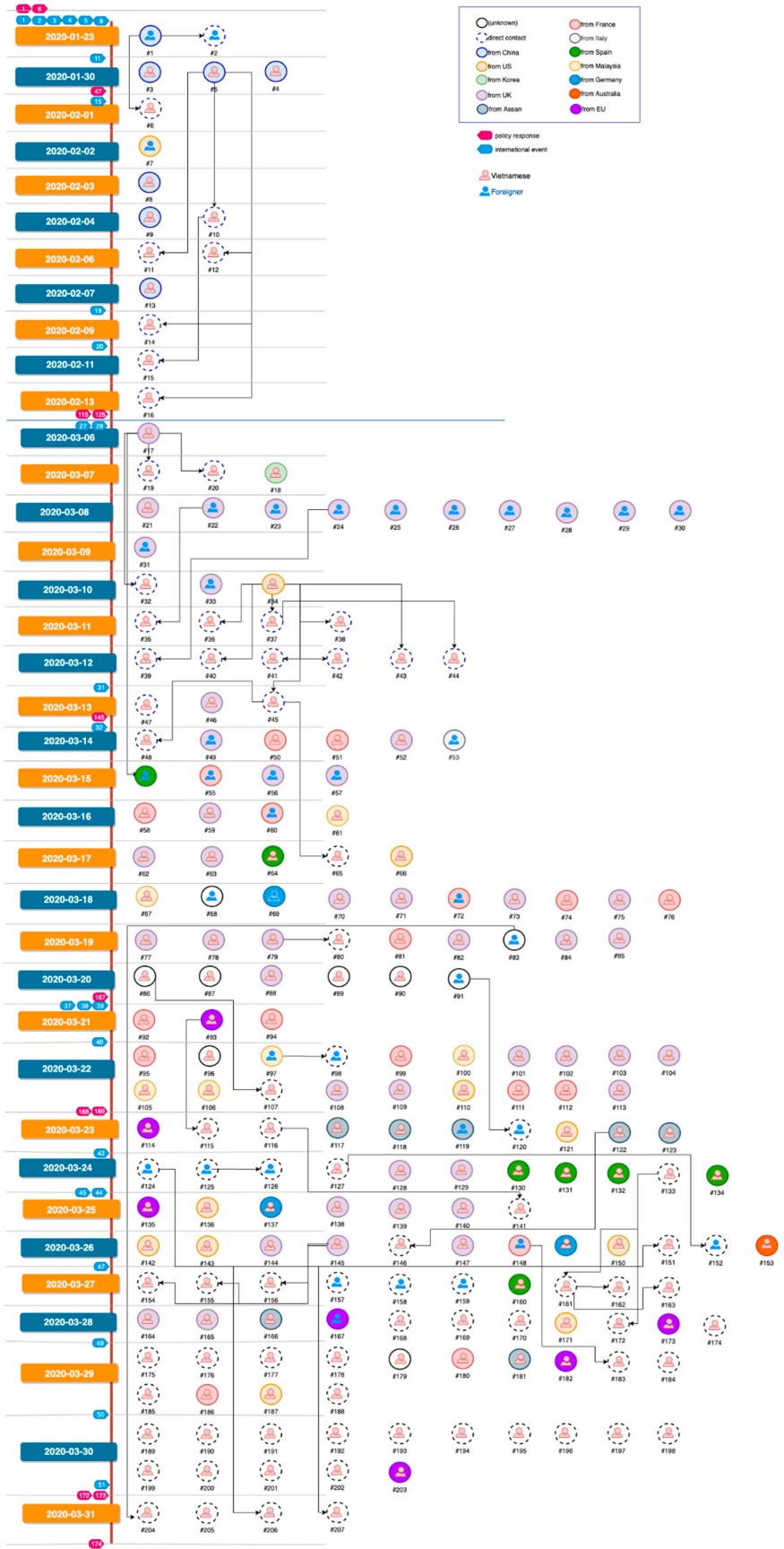

**Figure 3.** Chronology of COVID-19 in Vietnam (as of March 31, 2020).

Based on this timeline, four main periods of the COVID-19 outbreak in Vietnam are identified in Table 2.

**Table 2.** Four periods of the COVID-19 outbreak in Vietnam.

| Period | Date | Event |
|:---:|:---:|:---:|
| 1 | Before January 23 | No confirmed case in Vietnam |
| 2 | January 23–February 26 | First confirmed case in Vietnam–16th infected case discharged from hospital (Figure 4) |
| 3 | February 27–March 5 | No new case in Vietnam |
| 4 | March 6–present [April 4] | 17th infected case confirmed and more reported afterward |

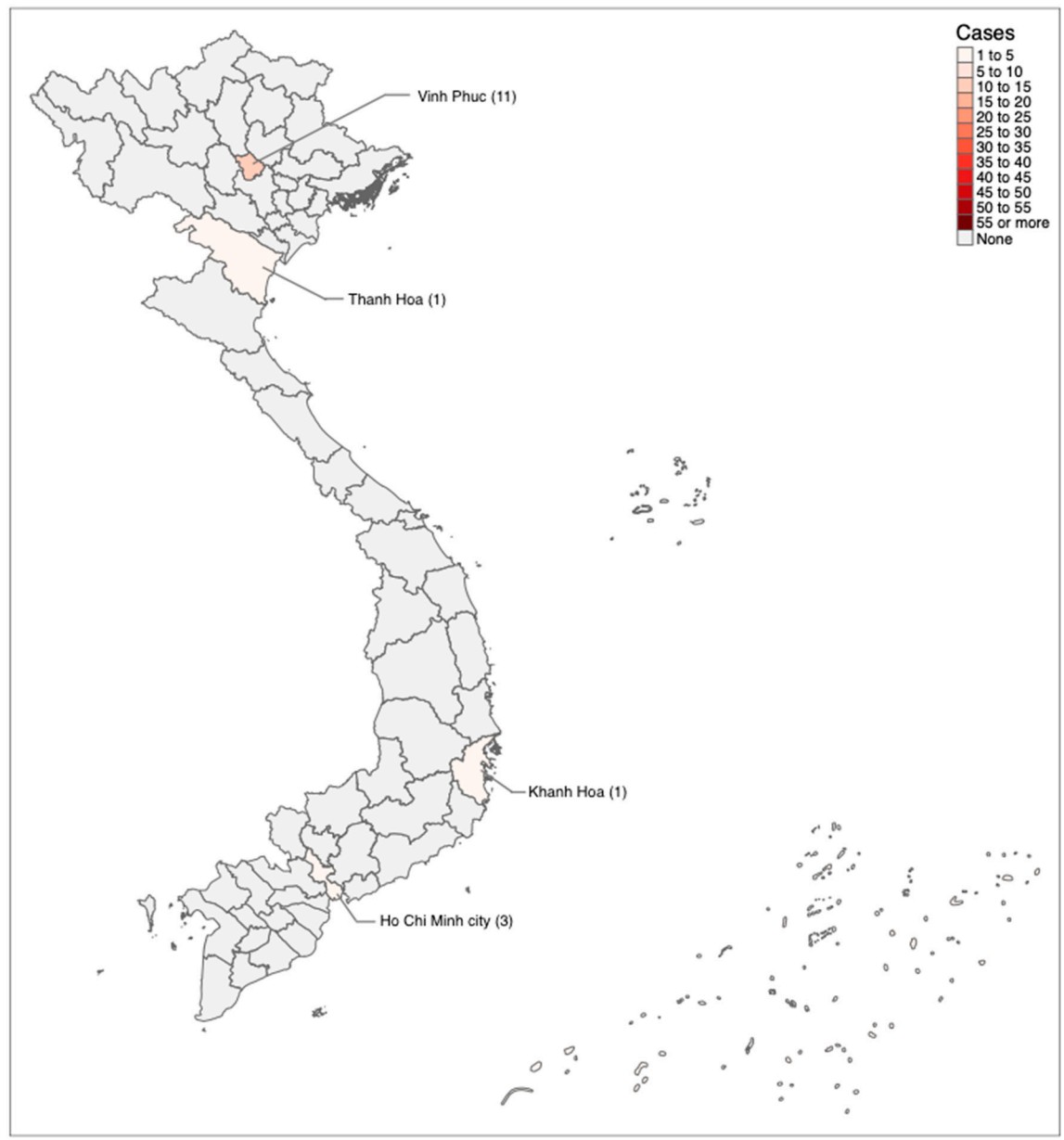

**Figure 4.** Locations of the first outbreak (16 cases) of COVID-19 in Vietnam.

## 4.2. Policy Response

We have manually extracted news reports regarding the government's actions following the outbreak. We have identified 173 official instructions, guidelines, plans, dispatch, policies and direct actions from the government, which were categorized as follows (Table 3):

**Table 3.** Categories of Vietnamese government policy response.

| Category | Count | Enacted in |
|---|---|---|
| Fake news prevention | 10 | Periods 2,4 |
| Assessment of the prevention | 5 | Periods 1,2 |
| Assessment of the threat | 4 | Periods 1,2 |
| Education | 15 | Periods 2,3,4 |
| Emergency response | 30 | Periods 2,4 |
| Guidelines and plans | 10 | Periods 1–4 |
| Innovation | 1 | Period 3 |
| Market control | 18 | Periods 2,3,4 |
| National funding | 1 | Period 4 |
| National Pandemic announcement | 1 | Period 4 |
| Outbreak-is-over announcement | 1 | Period 2 |
| Preventive action | 35 | Periods 2,3,4 |
| Reward | 1 | Period 1 |
| Social distancing announcement | 1 | Period 4 |
| Citizen support | 16 | Period 2,4 |
| Travel restrictions | 24 | Periods 2,3,4 |

### 4.2.1. Period 1: Before January 23, 2020

In this period before the first case was confirmed, the policy response focused on the assessment of the threat, together with developing guidelines and plans as preventive measures for the coming combat for the newly detected disease. As early as January 3, even before the first fatal case in China [12] and only a few days after China confirmed the outbreak of a new coronavirus [23], the Ministry of Health (MOH) had issued a directive on tightening quarantine at the Vietnam-China border [24]. On January 10, the Public Health Emergency Operation Center under the MOH followed up with a meeting to evaluate the disease situation and suggest preventive and treatment measures.

From January 16 to 20, the MOH issued two decisions (No. 125/QĐ-BYT and No. 156/QĐ-BYT) to provide guidelines and plans to prevent spread of the new coronavirus [25]. One urgent official dispatch (62/KCB-NV) to hospitals and local health departments stresses the importance of early disease prevention and detection [26].

### 4.2.2. Period 2: January 23, 2020–February 26, 2020

After the first confirmed case of COVID-19 infection, the policies focused both on minimizing risks from inbound travelers and containing the disease domestically. The policies include emergency responses, preventive actions, travel restrictions and market control.

During this period, because the hotspot of the disease was in China, attempts to control the spread from other countries focused on China-originated sources of infection. Vietnam started with strict screening on passengers from China at airports, seaports and land crossings, followed with isolating passengers suspected of infection, and entirely restricting flights to Wuhan and other affected areas in China [27]. On February 3 when the number of cases in China had shot up to 20,400, the Vietnamese

government heightened quarantine to cover all travelers who have come from or transited through the COVID-19 affected areas in China [28]. In the last days of the period, with Daegu, South Korea becoming the latest COVID-19 hotspot, the MOH added health declaration for all passengers from or transiting through South Korea and imposed isolation for those with symptoms [29].

At the same time, a series of emergency responses and preventive actions were made from the central to local governments. These included calls for increased cooperation among localities as well as specific tasks for multiple ministries and agencies [30,31]. With Deputy Prime Minister Vu Duc Dam appointed as the head of the steering committee for COVID-19 combat, intra-governmental cooperation is formalized and government officials are held to the highest accountability.

When the sixth COVID-19 case—also the first case of domestic transmission—was confirmed on February 1, the government declared the epidemic of a new coronavirus-caused infectious disease in Vietnam [32]. What followed were strict measures to prevent the virus from spreading, including quarantine, isolation of suspected virus carriers and voluntary isolation at the community [30]. This period was marked by a 20-day lockdown of a commune of 10,600 people in the northern province of Vinh Phuc after ten people tested positive to the new virus [33].

In response to potential public hoarding of certain goods, from February 1, the government had worked with relevant authorities to inspect pharmacies nationwide and withdraw business licenses of those which increased prices of face masks, hand sanitizers and medical gloves [34]. Within three days, more than 1200 drug stores were penalized and over 313,000 face masks were seized [35].

Another action taken by the government was the introduction of technological platforms, including the website http://ncov.moh.gov.vn and the NCOVI and Vietnam Health apps, to provide updated information about the epidemic, including testing data, advice on precautionary measures and live chat for questions related to COVID-19 [36].

However, there were two mistakes during this period. In the first case, the Vietnam National Administration of Tourism had attempted to promote a campaign called "Vietnam—Safe Haven" while the spread of COVID-19 was still at the early stage [37]. The campaign aimed to attract foreign tourists on the ground that Vietnam has managed this public health crisis well; consequently, 41% of the COVID-19 cases since patient 17—effectively a "patient zero" of the second cluster of the outbreak in Vietnam—are foreign tourists [38].

In the second case, the Ministry of Education and Training (MOET) had made three significant mistakes when school shutdowns began [39]. First, given the uncertain nature of the pandemic, it was a bad decision to close schools for one week at a time and then extend such shutdown week-by-week, leaving students and parents hanging every week and burdening their household decision-making. Second, because the ministry failed to provide educational guidelines timely, a longer break right after the Vietnamese national Tet holidays left the false impression of an extended vacation for families. This meant many families take the weeks off to travel, instead of staying home to minimize exposure and transmission. Third, the ministry's decisions on which groups of students (i.e., elementary and secondary) stay home and what universities should do (i.e., self-determination) were erratic and lacking scientific basis. When no age group has been proven "safe" from virus transmission, and young people could be asymptomatic carriers of the virus, letting them back to school could turn them into transmission vectors and threatened immunocompromised members of their own family, such as grandparents. Only after protest from the public did the ministry decree all school buildings to be shut down and all students to stay home.

Despite these few blunders, all confirmed cases were discharged from hospitals at the end of this period. Deputy Prime Minister Vu Duc Dam had declared that Vietnam "won the first battle against the epidemic" [40].

### 4.2.3. Period 3: February 27, 2020–March 5, 2020

Between February 26 and March 4, with no new patients detected, Vietnam entered a pause in the timeline of the outbreak. By contrast, the world, from Asia to Latin America and Sub-Saharan Africa, was seeing a rapid spread of the disease [41,42], with two new coronavirus hotspots emerging outside China. As of March 5, the number of infected patients in South Korea had hit 6284 and in Italy 3858 [3].

The pause in Vietnam, however, does not mean any changes to existing disease prevention and control policies. The government continued to impose stricter travel restrictions, such as halting visa exemption for citizens of severely affected countries, including South Korea and Italy, and ask schools and universities nationwide to keep closing. Pandemic combat remains a top priority. An army simulation exercise in response to the outbreak was held on March 4.

Yet, there were signs of imprudence. Deputy Prime Minister Vu Duc Dam, the leader of Vietnam National Steering Committee for COVID-19, had hastily announced on March 4 that: "If another week passes without new cases, Vietnam will announce the end of the epidemic" [43]. This leads to lax regulations among frontline guards against the disease and may have contributed to the outbreak after case 17.

### 4.2.4. Period 4: March 6, 2020–Current

After 22 days of having no new confirmed case of COVID-19, on March 6, the 17th patient was confirmed. This marked the second outbreak of Vietnam with the source of infection evaluated as much more complicated rather than just China or Korea (See Figure 5). The new cases in this period were mainly tourists and Vietnamese citizens coming back from European countries as well as people who had direct contact with the infected patients. On March 30, as the number of cases climbed every day, the Prime Minister declared the COVID-19 outbreak as a nationwide pandemic [44]. This declaration heightened alert over disease prevention, especially following the first cases of cross-infection to healthcare workers from a cluster at a tier-one hospital in Hanoi. As of March 31, there were 34 infected cases related to Bach Mai Hospital [45].

In response, the Vietnamese government imposed many rigorous measures [46], including a temporary suspension of visa issuance to all foreigners for 30 days effective on March 18 [47], obligated 14-day quarantine at centralized facilities from March 21 and ultimately, temporary suspension of entry to all foreigners on March 22 [48] (See Table 4).

**Table 4.** The halting of visa issuance to foreign countries.

| Country | Date Issued | Number of Cases in Vietnam (as of the announcement) |
|---|---|---|
| **All countries** | March 18 | 66 |
| **The United Kingdom and Schengen** | March 15 | 53 |
| **Italy** | March 3 | 17 |
| **South Korea** | February 29 | 16 |
| **China** | February 2 | 7 |

At the domestic level, the government has implemented urgent measures in multiple domains.

All schools have closed across the country and are likely expected to stay closed for a long period of time [49]. Large-scale quarantine and isolation of suspected cases persisted; and self-isolation for people with high risks of infection was carried out. All religious organizations are asked to stop holding mass gatherings from March 21 while all cultural, sports and entertainment activities in public places are prohibited from March 28 to April 15 [50,51]. The biggest change yet is, effective April 1, Vietnam enforced a "15-day nationwide social distancing" in which every household, village, commune, district and province would go into self-isolation [52]. Meanwhile, incoming flights to Vietnam are halted, while traveling within the country has also been restricted [53].

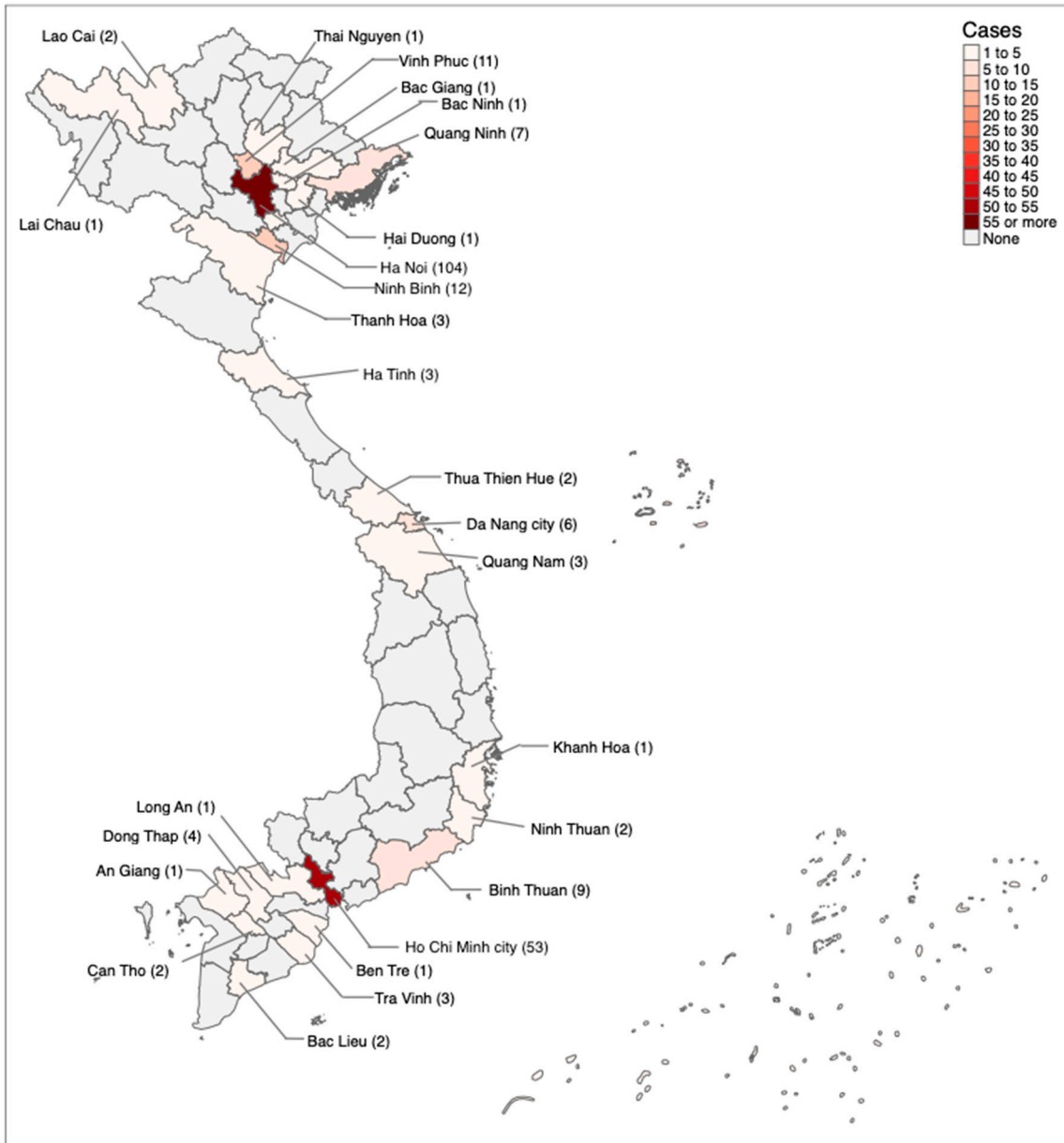

**Figure 5.** Locations of the second outbreak of COVID-19 in Vietnam (as of April 4, 2020).

Since the beginning of COVID-19 outbreak in Vietnam, the government has been focusing on hospital management policies to make sure the medical system is ready for the combat. In fact, the government issued several instructions about hospital management early on in late January, for example how to screen patients in hospital visits or specific distribution of responsibility to each level of hospital [54]. Since February, temporary hospitals have been set up to cure COVID-19 patients, for example the 300-bed field hospital in Vinh Phuc in the first outbreak [55] or two field hospitals to quarantine up to 1000 people in Hanoi [56]. From another perspective, two hospitals in Hanoi, namely Hong Ngoc and Saint Paul, used to be suspended or have a number of healthcare workers quarantined because of direct contact with COVID-19 patients without proper prevention measures. After these incidents, the city's authorities asked all medical centers to learn from these experiences and Hanoi Department of Health to conduct more effective training for healthcare workers as well as monitoring the process of taking care of infected patients to prevent the outbreak within hospitals [57]. In the second outbreak, the Vietnamese policy response related to hospitality management focused on Bach Mai Hospital as the biggest COVID-19 cluster in the country [58]. On March 28, Bach Mai hospital,

which is Hanoi's largest leading hospital, was completely blocked after twelve infected cases related to this medical center were confirmed. More than 5000 health care workers, patients and food workers in the hospital were tested for the virus and another 40,000 people who were at the hospital have been tracked down and asked to self-quarantine at home for 14 days [59]. During that night, a temporary hospital was set up within Bach Mai hospital to prepare for the worst scenario. However, two days later, the Ministry of Health announced that Bach Mai hospital cannot stop receiving and curing patients with serious and complex medical conditions from provincial hospitals, 80% of whom may lose their lives without proper treatment [60].

Overall, amid COVID-19 outbreak, the hospital management policies imposed by the government have been well followed by all-level hospitals—and as an encouragement—Prime Minister Nguyen Xuan Phuc acknowledged and praised the contribution of "soldiers in white blouse" in a letter sent to frontline doctors and nurses on March 26 [61].

Regarding economic activities, to assure the public over supply of essential goods, following the short panic-buying of Hanoi citizens when the 17th case was detected, officials across the country immediately worked with suppliers and distributors [62–64]. After the declaration of the nationwide social distancing on March 31, the government made sure to reassure people of the maintained transportation network for essential goods, especially food.

In addition, the State Bank of Vietnam, the country's central bank, on March 12 drafted a circular which would support credit organizations to restructure debt payment deadlines and cut borrowing interest rate and exemption for enterprises heavily affected by the pandemic. It was stated that over 44,000 customers with a total debt of VND222 trillion (US$9.51 billion) would benefit from this program [65]. Furthermore, on March 31, the government has discussed a welfare measure according to which all poor households would receive an aid of 1,000,000 VND per person per month [66].

In terms of technological application, the Vietnamese government opened a system to record electronic health declaration form for overseas travelers entering the country for the purpose of case monitoring and surveillance [67]. In addition, the Hanoi Smart City app was also activated to provide a risk assessment tool, consultation on prevention measurements, contact reports and live updates for Hanoi citizens [68].

The prompt and effective measures undertaken by the Vietnamese government to date have been highly regarded by international organizations [69]. Domestically, the results from global public research focusing on people's perception of their government's reaction by Dalia Research revealed that Vietnamese people have the most confidence in their government's response to the COVID-19 pandemic among 45 countries surveyed [70]. About 62% of the Vietnamese respondents think that the government is doing the "right amount' in response to the situation [71].

*4.3. Media Communication*

4.3.1. Official Press

Prior to the first case of COVID-19 in Vietnam, news on a "strange pneumonia" in China had circulated on Vietnamese media as early as the beginning of January 2020. Our dataset suggests news regarding this "strange pneumonia" first appeared on *Báo Chính phủ* (chinhphu.vn), *Vietnam Government Portal* and *Sức Khỏe và Đời Sống* [Health and Life] (suckhoedoisong.vn), the official news channel of the Ministry of Health, on January 9 [72,73].

According to the article written by the Public Health Emergency Operation Center on *Sức Khỏe và Đời Sống* [73], public health experts expected high risk of having an outbreak in Vietnam because it was near the Tet holidays (or Chinese New Year). The following preventive measures had been proposed:

- Monitoring information from WHO and other sources
- Communicating the information clearly to the citizen
- Increasing disease surveillance at the border

- Maintaining the alertness of the Public Health Emergency Operation Center and four Institutes of Hygiene and Epidemiology.
- Planning prevention and control measures.

It should be noted that other news outlets such as *Tuổi Trẻ, Thanh Niên or Quân Đội Nhân Dân* even shared the information to public earlier, from as early as January 3 [74,75]. Thus, the timely attention from newspapers and news media and afterward, social media, has played a crucial role in disseminating information to the public.

Since then, thousands of articles have been written updating Vietnamese people about the outbreak in the country and globally. Data from 14 online newspapers only generated nearly 15,000 articles published from January 9 to April 4. This helps considerably in raising public awareness about the disease as well as informing people on disease prevention and protection.

As can be seen from Figure 6, the amount of media communication to the public remained high at around 150 to 190 articles daily. However, there were days when newspapers appeared to pay attention to other events, and this number of articles dropped to below 100 in the third period. Overall, the flow of news and information to the public was on-going and rather substantial during the outbreaks.

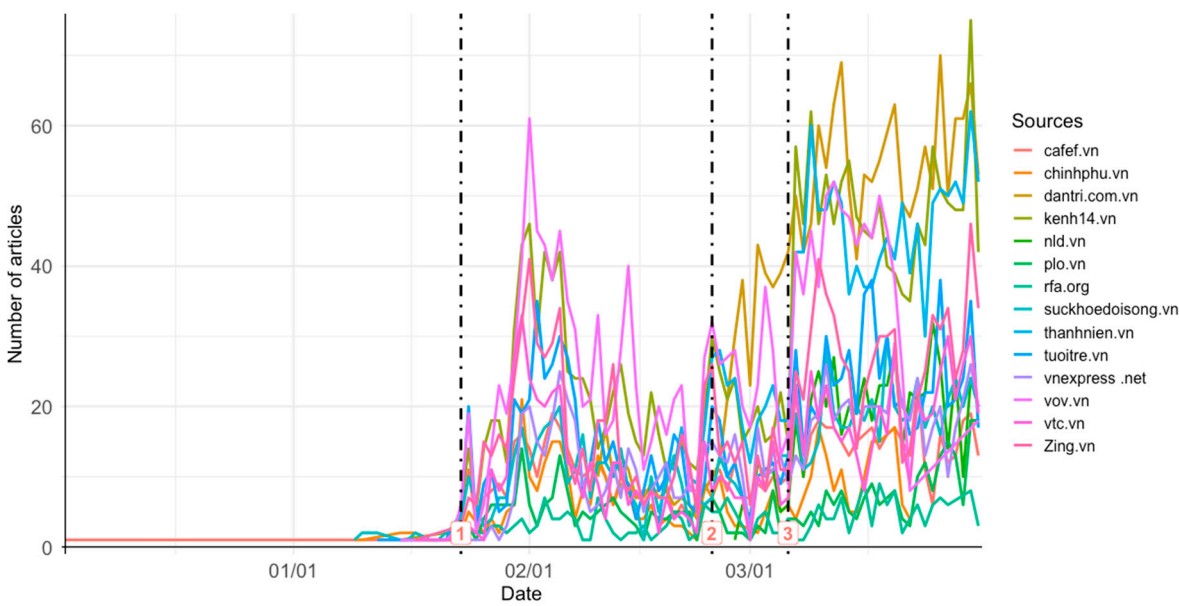

**Figure 6.** Number of articles about coronavirus in major newspapers.

4.3.2. Social Media

One particular characteristic of social media in Vietnam is the widespread use of Facebook (57.34% of the population [76]) and the local app Zalo (100 million users) [77]. These social media channels provide additional room for the government, particularly the Ministry of Health, to communicate coronavirus-related information to its citizens in a timely manner.

As Figure 7 suggests, the number of articles, including keywords such as Facebook or '*mạng xã hội*' (social media) in major Vietnamese newspapers is high. As the information has been disseminated through social media on an hourly basis, as well as the update from influencers or reporters, the newspapers had to remediate these contents. In the first and fourth periods, as many as 50 to 60 articles were circulated in each period. Meanwhile, 'Zalo' and '*cư dân mạng*' (netizen) received less attention.

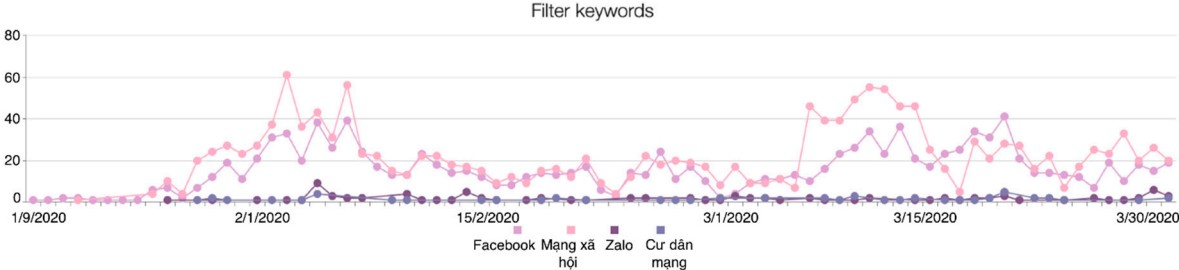

**Figure 7.** Number of articles about social media and coronavirus on major Vietnamese newspapers.

During the fourth period, information density proved to be an essential aspect of dealing with pandemic. There exists a stereotype of students and expatriates coming back from developed countries as being more informed and better behaved given their higher level of education. However, recent events have shown that stereotypes remain stereotypes. Patient 17 in Vietnam is a notable illustration: a well-educated upper-middle-class person who acted in an extremely ignorant manner, refused to self-quarantine for 14 days, and thus had likely spread the virus to many other cases afterward.

Media communication created both positive and negative impacts on public awareness and attention during this period. On the one hand, individual effort in supporting public awareness and protection against the virus was praised by social media. For example, Vietnamese dancer Quang Dang with the handwashing song, known as "*Ghen Co Vy*," went viral globally and attracted thousands of views, and many people around the world are posting their own covers [78]. Some Youtubers with a large number of subscribers have also contributed interesting and diverse perspectives about the pandemic, for instance the 16-year-old YouTuber named Melly Vuong [79]. In quarantine areas, people update about their life on social media, which spread a positive review of the facilities and healthcare system. For example, the well-known local fashionista Chau Bui made videos of her quarantine period, all of which amassed nearly one million views [80]. Famous artists are reported on social media for their donations to the healthcare facilities in Vietnam [81]. Social media also reacted vividly to cases of confirmed patients or quarantined people. For example, Patient 17, who returned from Europe and did not provide accurate health status declaration, got serious criticism from social media.

However, the negative side of rapid social media response is a strong emergence of fake news. The problem was most acute during the early days of critical events such as the confirmed case of the first or 17th patient with the involvement of celebrities and famous people. Responses to combat such mis/dis-information were made in both periods 2 and 4 (Table 3), formalized in a government decree in which anyone spreading fake news could be fined between (US$430-860), around 3–6 months' worth of basic salary in Vietnam [81].

### 4.3.3. Science Journalism

In the fight against coronavirus, science journalism plays a crucial role in informing the global research community as well as providing reliable information to the public. Vietnamese hospitals have contributed very promptly to the call for sharing knowledge and data about the disease [38] on January 28th, 2020, even before the call from Nature editors to all COVID-19 researchers [10]. Since then, there have been several significant works contributed by Vietnamese authors to the scientific community (see Table 5).

These efforts of Vietnamese scientists were also reported on the official press, for example, Vnexpress on January 30th posted an article entitled, "The first case of nCoV in Vietnam on the world's "medical bible"" [82], introducing the work by [38].

**Table 5.** Articles about COVID-19 published by Vietnamese authors.

| No. | Title | Authors | Journal/Source | Date |
|-----|-------|---------|----------------|------|
| 1 | Importation and Human-to-Human Transmission of a Novel Coronavirus in Vietnam | Phan, Lan T., Nguyen, Thuong V., Luong, Quang C., Nguyen, Thinh V. Nguyen, Hieu T., Le, Hung Q., Nguyen, Thuc T., Cao, Thang M. Pham, Quang D. | *The New England Journal of Medicine* | January 28, 2020 |
| 2 | Outbreak investigation for COVID-19 in northern Vietnam | Hai Nguyen Thanh, Truong Nguyen Van, Huong Ngo Thi Thu, Binh Nghiem Van, Binh Doan Thanh, Ha Phung Thi Thu, | *The Lancet Infectious Diseases* | March 4, 2020 |
| 3 | Duration of viral detection in throat and rectum of a patient with COVID-19 | Le Van Tan, Nghiem My Ngoc, Bui Thi Ton That, Le Thi Tam Uyen, Nguyen Thi Thu Hong, Nguyen Thi Phuong Dung, Le Nguyen Truc Nhu, Tran Tan Thanh, Dinh Nguyen Huy Man, Nguyen Thanh Phong, Tran Tinh Hien, Nguyen Thanh Truong, Guy Thwaites, Nguyen Van Vinh Chau | *medRxiv* | March 16, 2020 |
| 4 | The COVID-19 risk perception: A survey on socioeconomics and media attention | Toan Luu Duc Huynh | *Economics Bulletin* | March 25, 2020 |

Regarding initiatives of science journalism, the use of preprints to speed up the publishing process has been a focus in the combat against COVID-19 [83,84], for example [20]. This shows the positive signal of the Vietnamese scientific community to move forward with these global trends. The findings from Vietnamese hospitals and authors can, therefore, contribute to the database of global knowledge and expertise and may help to curtail this outbreak and prepare for future outbreaks.

On April 3, one of the leading universities in Vietnam, the National Economic University, also released a report titled "Assessment of COVID-19 impacts on the economy and policy recommendations," which has been heavily covered on the media [85]. Therefore, it appeared that researchers from hospitals and universities, rather than other types of institutions, have published more on COVID-19. This is consistent with the findings from [86] that university-affiliated authors in Vietnam tend to have higher research productivity than institution-affiliated peers.

Additionally, scientific publications and information also play a critical role in debunking myths on the disease and quickly communicating reliable information to the public, for instance, pushing back against a steady stream of rumors and conspiracy theories about the origin of the coronavirus outbreak [87]. This source may be used by newspapers and social media to disseminate information more widely to the general public (for example [88]). In Vietnam, various scientists have frequently updated scientific knowledge and perspectives on their personal Facebook accounts to inform the community. For instance, the Facebook posts by Tran Xuan Bach—an Associate Professor of John Hopkins University based in Hanoi—had attracted nearly 13,000 views and hundreds of shares from the public [89]. Furthermore, interviews with doctors and scientists were also conducted both in written form and in live form, for example, the live consultation session on the topic "Information about Coronavirus and respiratory diseases—How to prevent and treat" broadcasted on VTV News Newspaper [90].

In terms of scientific advancement, Vietnam has successfully produced test kits to diagnose COVID-19 infection in just one hour. The kit was stated to meet the WHO and the U.S. Center for Disease Control and Prevention Standards, and 20 countries were negotiating to buy these products [91]. Besides this, several scientific projects are being conducted to produce similar rapid testing kits to meet the increasing demand worldwide, for example, those by Hanoi University of Science and Technology

or Vietnam Academy of Science and Technology [91]. On March 3, the Vietnam Academy of Science and Technology officially announced the successful manufacturing of SARS-CoV-2 Virus Diagnostic Kit [92].

Besides scientific publications, university websites and various information-sharing magazines also contributed to enhancing the awareness of COVID-19. For instance, Phenikaa University has posted instructions to prevent the coronavirus up-to-date and donated more than 8000 liters of hand sanitizer to residential areas, Hanoi Youth Union, Departments of Education and hundreds of schools in Vietnam Northern provinces [93]. Similarly, more than 20 universities across the country have produced sanitizers and provided them free for their communities [94].

On March 31, 10 COVID-19 testing kiosks have been set up all around Hanoi, prioritizing the areas around Bach Mai hospital which was one of the largest recent outbreak hotspot [95]. Five thousand quick test kits—which give results after 10 minutes of testing—were distributed by the Ministry of Health to these testing kiosks, and were ready immediately on March 31st morning [96]. This is an important development, as Vietnam has so far relied on suboptimal methods such as self-report via declaration of medical status and targeting high-risk populations. Mass testing is a proactive measure that aims to directly inspect the entire population, leading to early detection and thus minimizing spread [97]. This was a lesson explicitly learned from South Korea, as not only the test kits were Korean-developed but the help of Korean experts was also enlisted in implementing mass testing in Vietnam. The importance of cross-border lesson learning underlines the significance of science journalism, especially when disease containment has to be coordinated on an international level.

### 4.3.4. Socioeconomic Aspects

While imposing strict directives to prevent the dissemination of the novel coronavirus, adverse impacts on the socio-economic situations in Vietnam are observable as consequences. For instance, the number of international visitors to Vietnam in the first three months of 2020 was expected to decline by 800,000 compared to last year [98]. However, as stated by Prime Minister Nguyen Xuan Phuc: "the government is willing to sacrifice economic benefits in the short term for the health of people," Vietnam government acknowledges the adverse consequences of preventive measures. The acknowledgment is also reflected through the simultaneous reduction of interest rate by the State Bank of Vietnam [99].

In the economic domain, for the stock market (Figure 8), before the detection of the first COVID-19 case in Vietnam, in period 1, the information for the disease in China appeared to have little impact on the market. During period 2, from January 23 to February 26, the market started the downward trend and lost more than 6.6%, from 959.58 points to 895.97 points. In period 3, the VN-Index experienced a minimal decrease from 895.97 on February 26 to 893.31 on March 5, 2020. During period 4, between confirmation of the 17th case in Vietnam on March 6 and the 207th case on March 31, the stock market suffered severely, with its benchmark VN-Index recording a sharp loss of nearly 229 points, or 25.7%.The weak market sentiment and force-sell pressure in Vietnam are predicted to persist, given the complicated developments in stock markets around the world.

While Vietnam is no longer a centrally planned economy [100,101], in the face of a public health crisis and national emergency as this one, the government was quick to control any sudden spikes in prices of consumer goods, and thus, effectively preventing price speculation and gouging. In the very first day of Period 4 of COVID-19 detection and intervention in Vietnam, there was an initial wave of panic stockpiling of food among local consumers [102]. However, within two days, the government met and discussed measures not only to cope with the disease outbreak but also to stabilize the domestic market [103]. By comparison, elsewhere around the globe, even in developed nations such as Australia [104] and the United States [105], consumers were reported to hoard a massive amount of food, toilet papers, hand sanitizer and anti-bacterial wipes; some were doing it so as to profit off the public's panic buying. While there is insufficient data for comparing prices before and after the outbreak among countries, it appears that the situation in Vietnam is kept relatively better under control than in

other countries. This is evidenced by the fact that the government, in conjunction with producers and supermarkets, was prompt to assure the public of food security as well as price stabilization.

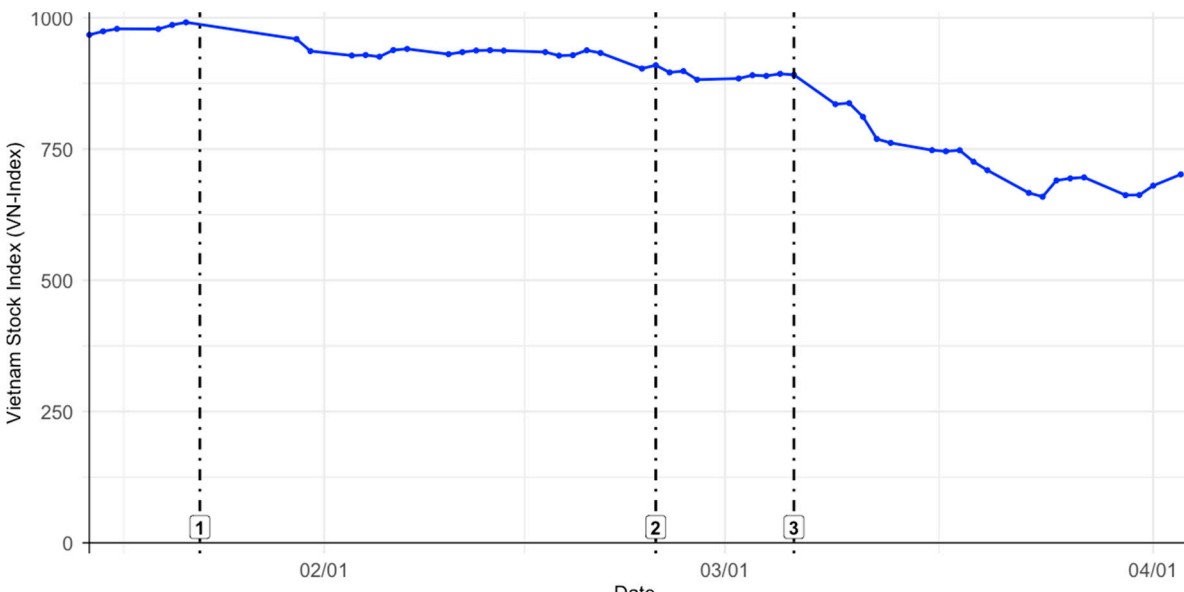

**Figure 8.** The changes in the benchmark VN-Index of Vietnam's stock market during the COVID-19 outbreak (As of April 4, 2020). Note: ① is the first case in Vietnam; ② is when the patient 16 recovered, ended the 1st outbreak; ③ is the 17th case in Vietnam, also the beginning of the 2nd outbreak.

In the social domain, during the early days after the first infected case, there had been rumors that the Vietnamese government was hiding information about the novel coronavirus, which caused some confusion and insecurity for the public [106]. To respond to this information, the authorities and mainstream media promptly reassured the citizens that transparency is the fundamental principle of the country in preventing the spread of the virus [107]. Government officials further explained that data and information from four Public Health Emergency Operation Centers of Vietnam were directly connected to the Centers for Disease Control and Prevention USA and, therefore, shared openly to the global database [107].

In the later periods, the emergence of strong political leaders such as Deputy Prime Minister Vu Duc Dam had a significant and positive influence on public perception of the Vietnamese government as well as the consensus and trust in Vietnam's efforts to fight against the pandemic. On social media, images and quotes of Deputy PM Dam appeared extensively on citizen's posts, which created a sense of solidarity and the belief in governments' efforts. Recently, the call from Prime Minister Nguyen Xuan Phuc on the whole nation's joint efforts in COVID-19 combat attracted public attention and support from individual citizens [108]. Responding to this call, one can easily see many images of bank transfer to the Vietnamese Fatherland Front on social media of Vietnamese people to support the government in the combat. On a larger scale, many enterprises, regardless of their size, also contributed to the national combat by donating their products such as masks, rice or milk, by donating their hotels for isolation wards—or most popularly—by donating cash [109].

The social response can also be seen from various groups, including residential, work-related and informal groups. Since the early days of the outbreak, residential groups played a central role in transferring information to individual citizens by means of public announcements, leaflets, posters or standees [110]. In big cities, apartment buildings as well as office buildings took measures to prevent the spread of the virus, for instance by sanitizing the whole building or checking people's temperatures before entering together with putting sanitizers in public spaces [111]. In daily transactions such as food shipping (see Figure 9), safety measures are being enforced by citizens. The image, once again,

reminds old people of how daily transactions were secretly conducted in the centrally planned economy period. In the digital space, multiple groups have been formed, especially on social media, to share information about the pandemic and collaborate efforts to help fight against it.

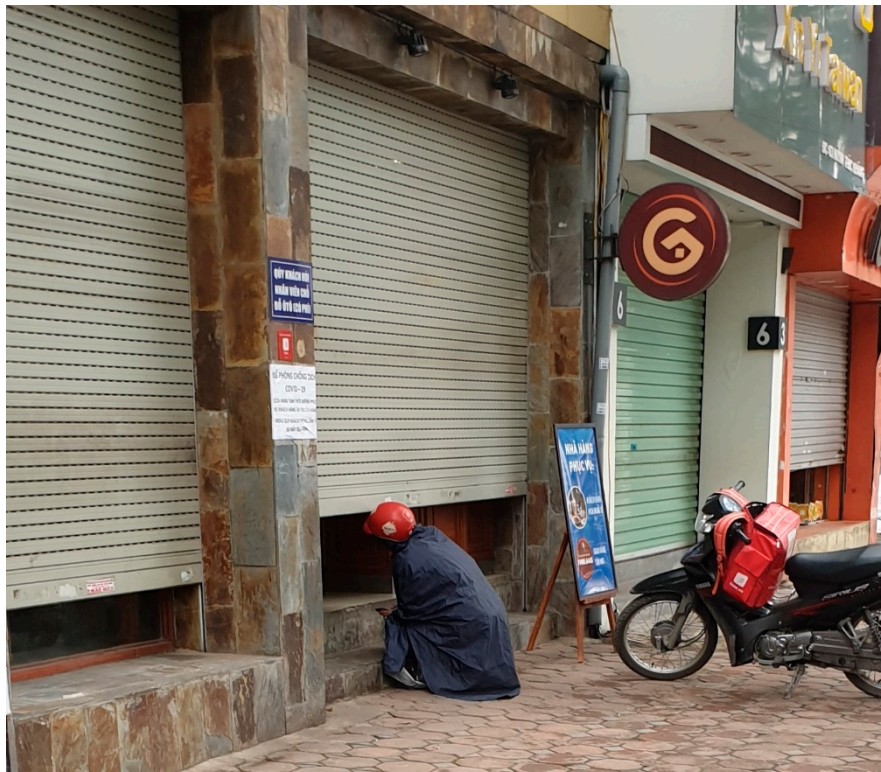

**Figure 9.** Safe business transaction in the COVID-19 time. ©2020 photo courtesy: Dam Thu Ha.

Several other social aspects of Vietnamese citizens had significant impacts on their reactions to the pandemic. Since the very early days, the Vietnamese people have paid close attention to the information through mass media and social media about the "strange pneumonia" from the neighboring country, China. This helped them to have preparedness and readiness to act very early on. This process of gradually distilling information also contributed to the fact that most of the time, the people responded to the situation without panicking. In addition, in the Vietnamese culture characterized by adoption of new ideas [112], the act of wearing masks has been seen as normal and as a way to protect oneself, not as a source of risks and infection for surrounding people as in some countries. Therefore, the habit of using face masks, after being encouraged by the government as an effective measure for protection, has been reinforced among the public.

**5. Discussion and Conclusions**

First of all, the early risk assessment and immediate action of the Vietnam government, as well as the seamless coordination between government and citizens, are some of the main contributors to the prompt and effective reaction vis-à-vis the COVID-19 pandemic, which is caused by the corona virus officially named SARS-CoV-2, in Vietnam up till now. To date, Vietnam has managed to keep the situation under control in several regards. All patients who tested positive either have recovered or were recovering; there has been zero death. In terms of prevention, the Vietnamese government has maintained rather impressive cooperation with citizens and measures such as mandatory mask-wearing, systemic health status declarations and checks and self-quarantine have all been swiftly and smoothly implemented. Despite having made certain missteps or near-missteps as have been analyzed in

the above sections, the government had been sensible enough to perceive warnings and recognize said mistakes.

The early policy response for preventive and treatment measures before the first case appearing in Vietnam is a highlight because Vietnam obtains a comparatively long shared borderline with China, and that was during the Tet holiday—the traditional new year of both Vietnam and China when the mobility rate was substantially high. The later simultaneous responses of the government presented through continuous directives of the Prime Minister according to the COVID-19 situation provide public health measures (school closure, public health quarantine, social distancing, etc.) and maintain the supply of fundamental goods for preventing the dissemination of the disease. Besides that, the effective control of the infected case number is also greatly influenced by the fluent coordination among governmental agencies. To achieve smooth national coordination during the harsh time, a whole-of-government pandemic prevention drill was held during the period when no newly infected cases were found. These efforts of Vietnam all met the suggestions by WHO for responding to community spread of COVID-19 [113] and implied the high awareness and integration of Vietnam government, which are two out of five main factors of a resilient health system proposed by [114]. Indeed, the government learned from its previous experience during the SARS 2003 epidemic and established a public health response mechanism that has proven to be effective to date.

Despite the optimistic results of the measures that have been taken to prevent the spread of COVID-19, one must not deny the mistake that may have intensified the severe consequences of the second outbreak. The Ministry of Education and Training was responsible for lack of guidelines for students as well as indecisive policies regarding school shutdowns during the early phases of the pandemic. The Ministry of Culture, Sports and Tourism had also taken missteps in underestimating the spread of the diseases and falling lax in controlling cross-borders contamination. It highlights a certain disconnect between the declaration of the PM to prioritize the health of citizens above economic concerns: indeed, the Ministry had thought to take advantage of a global pandemic to promote tourism, and Vietnam had paid dearly.

As has been suggested by Leach, *et al.* [115], governments often preferred to frame disease outbreaks as acute, thus relying on temporary, short-term measures of public intervention; an alternative to this framing would be to consider infectious diseases as endemic and long-present in the locality, thus adapting the entire community's lifestyle to deal with it. As COVID-19 is a novel disease, the Vietnamese government has indeed responded to it as an epidemic outbreak; however, it could also be observed that the rapid response itself owed to the fact that Vietnam had had a history of dealing with epidemics and pandemics, namely SARS in 2003 and H5N1 flu in 2008; as well as A/H1N1 in 2009, the disease continued to make its reappearance in smaller outbreaks in years to come, such as in 2018 [116]. For this reason, one may be able to have a positive outlook on Vietnam's sustainability, at least in terms of efficient decision-making, in the continuing battle against the spread of COVID-19.

Sustainability in terms of resources, however, merited a closer look. In view of the phenomenon of pre-lockdown panic-buying, particularly reported in the United States, but also in certain other countries such as France or Germany, food insecurity and commodity shortage have become legitimate concerns. In certain circles on social media, certain pictures of Vietnam and the US have been posted alongside to highlight the contrast between the quite adequate food distribution in Vietnam as well as the free meals provided in quarantine hospitals on one hand, and the empty shelves of not only food but also sanitary products (toilet rolls, hand sanitizer, soap, etc.) due to hoarding in certain places in the United States on the other. In addition, one could not discuss resource drainage in the face of a pandemic without evoking labor. It goes without saying that the medical personnel was on the frontline, but other than medical professions, it should be noted that manual laborers who uphold the infrastructure, such as store staff, garbage collectors, deliverers, etc. This aspect of resource merits as much attention from governments and planning as any other.

With regard to resources required to sustain prevention measures, there are positive signs. it is worth noting that Vietnam is efficient enough in the production of test kits for both domestic use

and export deals. This would be comparable to South Korea's testing ability, which had been put to use with remarkable success in curbing the spread of the disease as the country became the world's second-largest outbreak. [117]. However, as the second outbreak continued to spread, the efficiency of current preventive methods grew questionable. The demand for capable healthcare infrastructure to accommodate new cases thus remained pressing. Considering the fact that Vietnamese central hospitals suffered from chronic overpopulation in yearly minor outbreaks, this issue should very much concern policymakers. A suggestion would be to immediately devise plans to restrict mobility between provinces—both to prevent disease spread and to avoid overloading central hospitals—as well as to upgrade and equip regional hospitals and encourage infected citizens to utilize medical facilities in their proximity.

Concerning communication and information dissemination in the face of the pandemic, we have observed a pattern in the official press. Journalists have indeed picked up on the vocabulary used by officials in public speeches. As such, articles reporting on measures against COVID-19 employed rhetoric often associated to wars, such as: "fight the enemy" (*đánh giặc*, in which the word *giặc* connotes the illegitimacy of said enemy, a nuance difficult to translate), "leave nobody behind" (*không bỏ lại ai phía sau*; as if in a battle march), "grand solidarity" (*đại đoàn kết*, alluding to the two Indochina wars against France and the US), etc. This sort of highly combative language was, in fact, not new, as it has been used in official narratives for a good number of national media campaigns. On the other hand, technical terminologies seemed to be much less abundant in Vietnamese media reports.

Influential political leaders and experienced teams of officials were quick to recognize the crisis and implemented rigorous strategies to address the emerging outbreak. The media response has also helped in promoting public awareness about the disease and how people can protect themselves and the communities around them. Science journalism equally played a crucial role in communicating effectively and prompt information to the public and global research communities.

The three pillars of society's responses have contributed majorly to the situation of Vietnam, in which the community has responded quickly to a crisis and protect the interests of its citizens. It also reveals valuable lessons for other nations in the concurrent fight against the COVID-19 pandemic, namely an emphasis on mobilizing citizens' awareness of disease prevention without spreading panic, via fostering genuine cooperation between government, civil society and private individuals.

## 6. Limitations of the Study and Future Research Directions

We fully acknowledge the shortcomings of this paper. As the pandemic is still spreading and the situation continues to move rapidly at the time of writing, we are faced with a shortage of backing in the extant literature. While we have tried our best with rigorous methodology and technical tools—namely, using a web crawler to gather data *en masse* from news sites—there remain certain arbitrary choices that we had had to make as researchers, such as the choice of which news sites to consider, and which to exclude, from our analysis.

In terms of data curation and flexibility of analysis, our methodology also shows several limitations. It should be noted that news articles on online news sites were quite often reposted from one location to another, thus inflating the number of articles reporting on COVID-19 compared to the substantial information being disseminated. Rather than an error, we believe this phenomenon of media communication to be interesting in itself. However, we do not believe ourselves to be sufficiently equipped for analysis thereof, nor do we believe such an analysis would fit within the scope of our article. We would welcome any contribution, related or not to this point.

**Author Contributions:** conceptualization, V.-P.L. and Q.-H.V.; data curation, V.-P.L. and M.-H.N.; formal analysis, V.-P.L. and T.-H.P.; investigation, T.H.P., M.-T.H., K.-L.P.N., H.-K.T.N. and M.-T.H.; methodology, V.-P.L., M.-T.H., T.T. and Q.H.V.; project administration, M.-T.H.; resources, T.T. and Q.K.; software, V.-P.L. and Q.-H.V.; supervision, T.T. and Q.-H.V.; validation, M.-H.N., H.-K.T.N., M.-T.H., T.T., Q.K. and Q.-H.V.; Visualization, V.-P.L. and M.-T.H.; writing—original draft, T.H.P., K.-L.P.N. and T.-T.V.; writing—review & editing, T.H.P., M.-T.H., M.-H.N., K.-L.P.N., T.-T.V., H.-K.T.N., M.-T.H. and Q.K. All authors have read and agreed to the published version of the manuscript.

**Funding:** This research received no external funding.

**Acknowledgments:** We would like to dedicate this work to our homeland, Vietnam, to our government and to the Vietnamese people for all that we have done together as a nation in the combat against COVID-19. Our thoughts are with the people affected and all the brave and selfless healthcare workers who are helping those most in need. We hope that this difficult time for the whole humanity will end soon.

**Conflicts of Interest:** The authors declare no conflict of interest.

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
