# Peer review of "Policy Response, Social Media and Science Journalism for the Sustainability of the Public Health System Amid the COVID-19 Outbreak: The Vietnam Lessons"

_sustainability, doi:10.3390/su12072931_

Round 1
Reviewer 1 Report
An early risk assessment and immediate action of Public health are the bases for the emergency management. The study analyzes the situation in terms of Vietnam’s policy response, social media, and science journalism, using a self-made web crawl engine in order to scan and collect official media news related to COVID-19. The results reports data regarding the local situation of Vietnam, that can be biased by the lack of a precise disease diagnosis; nevertheless, the authors carry on an interesting exercise of event based surveillance. Although the limits of the study, in my opinion the manuscript is suitable for publication in an international impacted journal.
At the same time, the manuscript is too long and difficult to read. I suggest to shorten, deleting some tables and figures (e.g. tables 1 and figure 2). Aims of the study should be better explained, and a professional English editing performed.
Author Response
Dear reviewer 1,
Once again we would like to express our most sincere thanks for your contribution in helping us improve our submission. We have done major revisions following your very detailed input. The changes brought to the manuscript have been highlighted in yellow, whereas newly added passages were highlighted in green. In this letter, we will address each of the points you have made individually.
- An early risk assessment and immediate action of Public health are the bases for the emergency management. The study analyzes the situation in terms of Vietnam’s policy response, social media, and science journalism, using a self-made web crawl engine in order to scan and collect official media news related to COVID-19. The results reports data regarding the local situation of Vietnam, that can be biased by the lack of a precise disease diagnosis; nevertheless, the authors carry on an interesting exercise of event based surveillance. Although the limits of the study, in my opinion the manuscript is suitable for publication in an international impacted journal.
Thank you for your comments which helped us a lot in revising our work and making it more concise as well as useful for both academic and general audience.
- At the same time, the manuscript is too long and difficult to read. I suggest to shorten, deleting some tables and figures (e.g. tables 1 and figure 2). Aims of the study should be better explained, and a professional English editing performed.
We have edited the manuscript thoroughly according to your suggestion. Firstly, we have cut down the total length of the paper to 27 pages, with 11,424 words, 9 figure and 5 tables. Table 1 and Figure 2 in the previous version were deleted.
Aims of the study are explained in detail in the Introduction section as follows:
While Vietnam also saw an uptick in new cases during March, its response to COVID-19, which is a combination of political readiness, timely communication, and scientific journalism, offers valuable lessons in dealing with situations of epidemics on a state level. As of March 31, the number of COVID-19 infections in Vietnam was 207, in which 58 patients had recovered, 149 are being treated, and no deaths recorded (See Figure 1).
The study, though in its preliminary and subject to changes as the disease progresses, might nonetheless be instructive and helpful for other countries to better understand the role of policy response, social media, and science journalism in maintaining public health. The case of Vietnam provides empirical evidence for assessing the efficacy of specific measures in fighting the pandemic.
Moreover, toward the end of the Literature Review, we have also emphasized the aims of the study:
The few studies on the outbreak in Vietnam have largely focused on the clinical aspects [20], [21], [22] with the exception of [23], which is on public risk perception. This piece analyzes the government’s response in terms of public health measures and policy implementation, as well as the mobilization of citizens’ collaboration in containing the disease have been very limited. This shall be a subtle call to action for researchers in Vietnamese social sciences.
Finally, the manuscript has been edited thoroughly to make sure there is no language mistake left.
We believe that we have put an appropriate amount of efforts in responding to all of your concerns. We hope that you find our revision satisfactory, and would like to once again thank you for your suggestions. We are truly honored to be able to work with you in improving this manuscript.
With our highest respects,
The authors
Reviewer 2 Report
This is a timely and interesting paper on the Vietnamese response to COVID 19 outbreak.
The authors honestly acknowledge the shortcomings of their paper. While it provides interesting and useful informations both for professionals and general audience it indulges in long descriptions of the work done by Vietnam authorities, scientists and scientific journalists in a far more detailed manner than is usual for a scientific audience who would appreciate a shorter and more concise text.
In my opinion the paper could be much more effective if the text is shortened and some non essential information removed.
This is particularly true for chapter 2 on literature review, chapter 4 on policy response, and 4.3.3 on science journalism and 4.3.4 on socioeconomic aspects
Author Response
Dear reviewer 2,
Once again we would like to express our most sincere thanks for your contribution in helping us improve our submission. We have done major revisions following your very detailed input. The changes brought to the manuscript have been highlighted in yellow, whereas newly added passages were highlighted in green. In this letter, we will address each of the points you have made individually.
- This is a timely and interesting paper on the Vietnamese response to COVID 19 outbreak. The authors honestly acknowledge the shortcomings of their paper. While it provides interesting and useful informations both for professionals and general audience it indulges in long descriptions of the work done by Vietnam authorities, scientists and scientific journalists in a far more detailed manner than is usual for a scientific audience who would appreciate a shorter and more concise text.
Thank you for your comments which highlighted the main aims of this study. The sharing of lessons learnt among nations and within the research community is especially imperative when the situation becomes rapidly worsening and complicated worldwide. Therefore, we attempt to provide comprehensive insights of Vietnam’s efforts in terms of policy response, social media and science journalism based on the data collected.
- In my opinion the paper could be much more effective if the text is shortened and some non essential information removed. This is particularly true for chapter 2 on literature review, chapter 4 on policy response, and 4.3.3 on science journalism and 4.3.4 on socioeconomic aspects.
Thank you for your comments. We have revised the manuscript heavily to cut down to the length to 27 pages, with 11,424 words, 9 figure and 5 tables.
Particularly, In the literature part, we shortened the text by cutting down information about publishers' attempts in opening free resources as well as the unnecessary information about some previous studies in other countries.
In policy response, we summarized key actions and announcements by the government and updated major moves since the submission date (March 22 - now).
In science journalism part, we cut off table 7 and the part explaining studies from Vietnam in the topic of COVID-19. We also highlighted the role of science communication in the recent act of Hanoi authorities to apply lesson learnt of large-scale testing from South Korea.
Last but not least, the socioeconomic aspect is now shorter as we deleted unnecessary information such as table 8 - the price of commodities in Vietnam.
Even though a large part of unnecessary text and information were edited. We were still able to update new information and statistics as of March 31. Given urgent need to understand the COVID-19 responses across the world, the timely and much-needed update of data and statistics from Vietnam hope to provide new and latest insights to the public.
We believe that we have put an appropriate amount of efforts in responding to all of your concerns. We hope that you find our revision satisfactory, and would like to once again thank you for your suggestions. We are truly honored to be able to work with you in improving this manuscript.
With our highest respects,
The authors
Reviewer 3 Report
It is clearly understood that the situation of this pandemic is changing very frequently. It is really appreciable that the way Vietnam managed to contain it so far. Throughout the manuscript, one specific aspect was not really emphasized how the role of research organizations, hospital managements worked in coherence to successfully contain it. Explaining this might give further broader picture which could be an example for many other countries to follow.
Figure qualities (e.g. figure 7) can be improved further.
Author Response
Dear reviewer 3, Once again we would like to express our most sincere thanks for your contribution in helping us improve our submission. We have done major revisions following your very detailed input. The changes brought to the manuscript have been highlighted in yellow, whereas newly added passages were highlighted in green. In this letter, we will address each of the points you have made individually. • It is clearly understood that the situation of this pandemic is changing very frequently. It is really appreciable that the way Vietnam managed to contain it so far. Throughout the manuscript, one specific aspect was not really emphasized how the role of research organizations, hospital managements worked in coherence to successfully contain it. Explaining this might give further broader picture which could be an example for many other countries to follow. Thank you for your insightful comment. We have provided more information relating to how research organizations and hospital managements contribute to the fight against COVID-19. Surprisingly, universities have been more proactive than research organizations in producing COVID-19-related results: On April 3, one of the leading universities in Vietnam, National Economic University, also released a report named “Assessment of COVID-19 impacts on the economy and policy recommendations”, which has been heavily covered on media [84]. Therefore, it can be observed from recent moves in the Vietnamese scientific community that researchers from hospitals and universities, rather than other types of research organizations appeared to publish more actively in the global search for latest scientific knowledge on COVID-19. This is consistent with the findings from [85] that contrary to the usual belief, university-affiliated authors in Vietnam turned out to have higher research productivity than institution-affiliated peers. Meanwhile, hospitals haves strictly followed the government’s guidelines and instructions. There are cases in which hospitals were infected, but swift responses and authoritative managements have put the situations under control: Since the beginning of COVID-19 outbreak in Vietnam, the government has been focusing on hospital management policies to make sure the medical system is ready for the combat. In fact, the government issued several instructions about hospital management early on in late January, for example how to screen patients in hospital visits or specific distribution of responsibility to each level of hospital [54]. Since February, temporary hospitals have been set up to cure COVID-19 patients, for example the 300-bed field hospital in Vinh Phuc in the first outbreak [55], or two field hospitals to quarantine up to 1,000 people in Hanoi [56]. From another perspective, two hospitals in Hanoi, namely Hong Ngoc and Saint Paul, used to be suspended or have a number of health-care workers quarantined because of direct contact with COVID-19 patients without proper prevention measures. After these incidents, the city’s authorities asked all medical centers to learn from these experiences and Hanoi Department of Health to conduct more effective training for health-care workers as well as monitoring the process of taking care of infected patients to prevent the outbreak within hospitals [57]. In the second outbreak, the Vietnamese policy response related to hospitality management focused on Bach Mai Hospital as the biggest COVID-19 cluster in the country [58]. On March 28, Bach Mai hospital, which is Hanoi’s largest leading hospital, was completely blocked after twelve infected cases related to this medical center were confirmed. More than 5,000 health care workers, patients and food workers in the hospital were tested for the virus, and another 40,000 people who were at the hospital have been tracked down and asked to self-quarantine at home for 14 days [59]. During that night, a temporary hospital was set up within Bach Mai hospital to prepare for the worst scenario. However, two days later, the Ministry of Health announced that Bach Mai hospital cannot stop receiving and curing patients with serious and complex medical conditions from provincial hospitals, 80% of whom may lose their lives without proper treatment [60]. Overall, amid COVID-19 outbreak, the hospital management policies imposed by the government have been well followed by all-level hospitals, and as an encouragement, Prime Minister Nguyen Xuan Phuc acknowledged and praised the contribution of “soldiers in white blouse” in a letter sent to frontline doctors and nurses on March 26 [61]. • Figure qualities (e.g. figure 7) can be improved further. We have improved the figure qualities. Moreover, we also submitted a high quality version of the figures alongside the manuscript, which will be used in the later process. We believe that we have put an appropriate amount of efforts in responding to all of your concerns. We hope that you find our revision satisfactory, and would like to once again thank you for your suggestions. We are truly honored to be able to work with you in improving this manuscript. With our highest respects, The authors